# Epigenome-wide meta-analysis of PTSD across 10 military and civilian cohorts identifies methylation changes in *AHRR*

Alicia K. Smith ⓘ et al.#

Epigenetic differences may help to distinguish between PTSD cases and trauma-exposed controls. Here, we describe the results of the largest DNA methylation meta-analysis of PTSD to date. Ten cohorts, military and civilian, contribute blood-derived DNA methylation data from 1,896 PTSD cases and trauma-exposed controls. Four CpG sites within the aryl-hydrocarbon receptor repressor (*AHRR*) associate with PTSD after adjustment for multiple comparisons, with lower DNA methylation in PTSD cases relative to controls. Although *AHRR* methylation is known to associate with smoking, the *AHRR* association with PTSD is most pronounced in non-smokers, suggesting the result was independent of smoking status. Evaluation of metabolomics data reveals that *AHRR* methylation associated with kynurenine levels, which are lower among subjects with PTSD. This study supports epigenetic differences in those with PTSD and suggests a role for decreased kynurenine as a contributor to immune dysregulation in PTSD.

#A list of authors and their affiliations appears at the end of the paper.

Post-traumatic stress disorder (PTSD) is characterized by re-experiencing, avoidance, and hyperarousal symptoms that can negatively impact mood and physiologic health[1]. Although not everyone who experiences trauma goes on to develop PTSD, those who do often experience severe and disabling symptomatology[2,3]. Research suggests that both genetic and environmental factors contribute to risk for developing PTSD[4,5].

Given the necessary, but not sufficient, role of environmental exposure (i.e., trauma) in the development of PTSD, it is critical to characterize the pathways underlying differential risk and resilience. Studies that contextualize the role of environmental influences provide additional insight into modifiable factors that may promote post-trauma resilience. For example, lack of social support at the time of trauma is associated with increased risk of developing PTSD in both military and civilian contexts[6]. Similarly, studies that investigate how presumed environmental influences might affect biological pathways could provide insights into the genes whose regulation patterns differ in those with PTSD. A growing line of research aimed at elucidation of mechanisms by which environmental factors contribute to PTSD, has focused on epigenetic modifications, which regulate gene function in response to environmental triggers. Epigenetic modifications, such as DNA methylation at cytosine-guanine dinucleotides (CpG sites), can induce changes in gene expression that are maintained through each round of cell division.

Multiple reviews have linked traumatic stress to differences in the proportion of methylated DNA intensity to non-methylated DNA intensity (ß) at specific CpG sites[7–10]. Indeed, a number of epigenome-wide association studies (EWAS) of individual cohorts have identified PTSD-associated CpG sites in genes and pathways related to neurotransmission and immune function[11–15]. Similarly, studies using methylation of specific CpG sites that predict chronological age with reasonably high accuracy to capture age acceleration (deviations from a line of prefect prediction), demonstrate associations with PTSD and link differences in peripheral DNA methylation to memory formation and neural integrity[16,17]. Although promising, the extant literature of EWAS studies on PTSD is limited by use of individual cohorts with small sample sizes and varying analytic pipelines that can make it challenging to synthesize findings. Consortia efforts can overcome these limitations by providing a shared analytic pipeline and increasing sample size and thus statistical power. The goal of this study is to capitalize on consortium strengths by conducting a meta-analysis of DNA methylation across 10 military and civilian cohorts participating in the Psychiatric Genomics Consortium (PGC) PTSD Epigenetics Workgroup. Our results suggest that lower aryl-hydrocarbon receptor repressor (AHRR) methylation in those with PTSD correlates with lower kynerunine levels, which may contribute to immune dysregulation in PTSD.

## Results
**Participating cohorts**. Sample characteristics for the 10 cohorts that have contributed data are listed in Table 1 ($N = 1896$). All participants were exposed to trauma, and 42% had a current diagnosis of PTSD. There were no significant age differences in PTSD cases and trauma-exposed controls. However, the demographic characteristics for each cohort varied substantially (Supplementary Data 1), with cases more likely to be male ($X^2 = 22.4$; $p < 0.05$) and current smokers ($X^2 = 54.2$; $p < 0.05$) across the majority of cohorts and in the overall cohort.

**PTSD-associated CpG sites from Meta-Analysis**. In our primary meta-analysis, we found 4 CpG sites associated with current PTSD after correction for multiple comparisons ($-6.60 < z < -5.57$; $p < 3.6E-08$; Fig. 1; Table 2; Supplementary Data 2). AHRR contains the

top 4 PTSD-associated CpGs, with lower methylation in PTSD cases relative to controls (Supplementary Fig. 1). This was consistent when stratified by both sex (Supplementary Fig. 2) and ancestry (Supplementary Fig. 3). Data from iMETHYL, which reports eQTMs associated at false-discovery rate (FDR) < 0.05, suggests that methylation of both cg05575921 and cg25648203 are inversely associated with AHRR expression in peripheral blood mononuclear cells (PBMCs).

**Sensitivity analysis with smoking status**. As lower methylation of AHRR CpG sites has been associated with smoking[18–20], we controlled for smoking status in our sensitivity analyses of the 4 significant CpGs (Supplementary Fig. 4). The association between AHRR methylation and PTSD was attenuated for all four CpGs. Since there is a higher rate of smoking among PTSD cases, we evaluated potential differences of effect between PTSD and smoking status by testing the associated CpGs separately in smokers and non-smokers (Fig. 2) and by plotting the associated CpGs separtely by smoking status (Supplementary Fig. 5). The associations between AHRR CpGs and PTSD were most prominent in non-smokers. Of the multiple stratified analyses performed (Supplementary Figs. 2, 3, and 5), the effect sizes were consistent across different strata, with the smokers being the only one did not show an association with PTSD.

We further delineated the association between PTSD and smoking-associated CpG sites from a large meta-analysis on smoking conducted by the CHARGE consortium[20]. We hypothesized that unreported smoking among those that identify as non-smokers would result in the appearance of an association between smoking-associated CpG sites and PTSD. This evaluation included 21 CpGs representing the most significantly associated CHARGE smoking sites and CpGs in AHRR. If the association between AHRR and PTSD was due to unreported smoking in PTSD controls, we would expect to see a similar effect across all smoking-CpGs. However, after adjusting for the number of tests performed, AHRR CpG sites remained associated with PTSD in the non-smokers only, and no smoking CpG showed a significant association with PTSD in the controls or cases. These findings are consistent with the hypothesis that the association between PTSD and DNA methylation of AHRR is independent of smoking status (Supplementary Fig. 6).

**Tryptophan catabolism in PTSD**. We next evaluated other biological factors that could contribute to aryl-hydrocarbon receptor (AHR) expression using linear regression models. AHR is activated in many immune cell types, including T cells, B cells, and NK cells, by indolamine-mediated tryptophan catabolism[21]. Inflammation can shift tryptophan metabolism away from serotonin synthesis towards kynurenine synthesis[22]. To evaluate the association of tryptophan catabolism in PTSD and its relationship to AHRR methylation, we leveraged data from the 116 subjects from MRS that had both DNA methylation and tryptophan metabolite data. In this group, we first note that kynurenine levels were lower in subjects with PTSD relative to controls ($t = -2.00$; $p = 0.048$; Supplementary Fig. 7). Consistent with this observation, lower methylation of AHRR CpGs associated with lower kynurenine (cg21161138; $t = 2.09$, $p = 0.039$) and lower kynurenic acid (cg05575921; $t = 2.95$, $p = 0.004$).

Further, metabolome data in the MRS confirmed that self-reported smoking status is consistent with empirical cotinine levels; 87% of self-reported non-smokers and light smokers had cotinine levels consistent with no-smoking, second-hand smoking or light exposure, while 91% of regular or heavy smokers had cotinine levels consistent with their endorsement. In addition, these data also provide insight into why controlling for smoking may attenuate the association between DNA methylation and PTSD. We noted an inverse relationship between cotinine and

**Table 1 Overview of the cohorts.**

| Study | N | PTSD (%) | Male (%) | Smokers (%) | White (%) | Black (%) | Hispanic (%) | Other or unknown (%) |
|---|---|---|---|---|---|---|---|---|
| DNHS | 100 | 40% | 40% | 32% | 15% | 85% | 0% | 0% |
| GTP | 265 | 28% | 29% | 30% | 6% | 94% | 0% | 0% |
| WTC | 180 | 47% | 100% | 9% | 76% | 4% | 0% | 20% |
| Army STARRS | 102 | 50% | 100% | 70% | 100% | 0% | 0% | 0% |
| MRS | 126 | 50% | 100% | 56% | 57% | 8% | 25% | 9% |
| INTRuST | 303 | 38% | 66% | 25% | 67% | 19% | 7% | 7% |
| PRISMO | 62 | 52% | 100% | 61% | 100% | 0% | 0% | 0% |
| VA-M-EA | 176 | 49% | 78% | 26% | 100% | 0% | 0% | 0% |
| VA-M-AA | 369 | 50% | 50% | 30% | 0% | 100% | 0% | 0% |
| VA-NCPTSD | 213 | 69% | 90% | 23% | 100% | 0% | 0% | 0% |
| Civilian Subtotal | 545 | 42% | 54% | 23% | 31% | 63% | 0% | 7% |
| Military Subtotal | 1351 | 42% | 56% | 34% | 61% | 32% | 4% | 3% |
| Total | 1896 | 42% | 55% | 31% | 52% | 41% | 3% | 3% |

Participating civilian cohorts include: Detroit Neighborhood Health Study (DNHS), Grady Trauma Project (GTP), World Trade Center 9/11 Responders. Participating military cohorts include: SUNY (WTC), Army Study to Assess Risk and Resilience in Servicemembers (Army STARRS), Injury and Traumatic Stress (INTRuST), Marine Resiliency Study (MRS), Prospective Research in Stress-related Military Operations (PRISMO), Mid-Atlantic VA VISN 6 MIRECC (VA-M-AA & VA-M-EA), Boston VA/National Center for PTSD (VA-NCPTSD).

**Table 2 CpG sites significantly associated with current PTSD.**

| CpG | Location | Gene | Features | Z | p-value |
|---|---|---|---|---|---|
| cg05575921 | chr5:373378 | AHRR | Body | −6.60 | 4.00E-11 |
| cg21161138 | chr5:399360 | AHRR | Body | −6.26 | 3.94E-10 |
| cg25648203 | chr5:395444 | AHRR | Body | −5.77 | 8.06E-09 |
| cg26703534 | chr5:377358 | AHRR | Body | −5.57 | 2.50E-08 |

Results of the EWAS meta-analysis of ten cohorts ($N_{cases} = 878$, $N_{controls} = 1018$). Association analyses of each cohort are based on empirical Bayes method. Meta-analysis is done by sample size-weighted sum of z-scores. p-values are two-sided and unadjusted for multiple testing. All of the association models are adjusted for sex (if applicable), age, CD8, CD4, NK, B cell, and monocyte cell proportions, and ancestry using principal components (PCs).

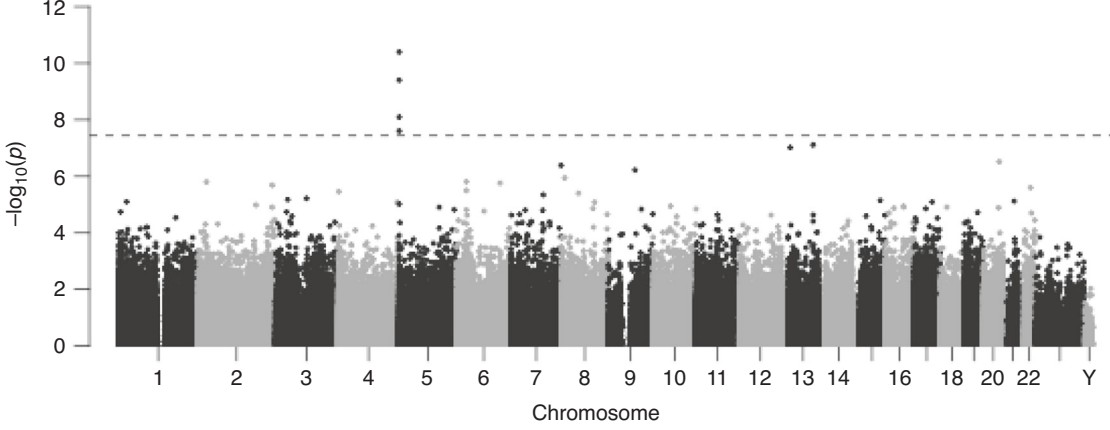

**Fig. 1 PTSD associates with DNA methylation differences across the genome. a** Manhattan plot for the meta-analysis of 10 cohorts ($N_{cases} = 878$, $N_{controls} = 1{,}018$). Association analyses of each cohort are based on empirical Bayes method. Meta-analysis is done by sample size-weighted sum of z-scores. p-values are two-sided and unadjusted for multiple testing. All of the association models are adjusted for sex (if applicable), age, CD8, CD4, NK, B cell, and monocyte cell proportions, and ancestry using principal components (PCs). The x-axis is the location of each site across the genome. The y-axis is the −log10 of the p-value for the association with PTSD. The dashed line indicates statistical significance at $p < 3.6E-8$.

kynurenine ($r = -0.267$; $p = 0.0004$) that was consistent among both smokers and non-smokers (Supplementary Fig. 8).

## Discussion

An individual's risk of developing PTSD depends on both the nature of the trauma and the physiological response to that trauma[23]. Not all individuals exposed to trauma develop PTSD, and a better understanding of the modifiable biological factors underlying risk and resilience will inform the development of new prevention and treatment strategies. In this study we identified CpG sites associated with PTSD. We observed that, on average, PTSD cases had lower methylation at several CpG sites in the AHRR gene when compared to trauma-exposed controls. Methylation of AHRR CpGs has been strongly linked to smoking[18–20]. As substantially more of the PTSD cases reported smoking compared to controls, this suggested that we should control for smoking in the EWAS. A comparable approach was taken by Marzi and colleagues[24]. In their study of DNA methylation in relation to victimization stress in adolescents, they noted experiment-wide significant associations in 3 of the 4 AHRR CpGs associated with PTSD. Similar to our findings,

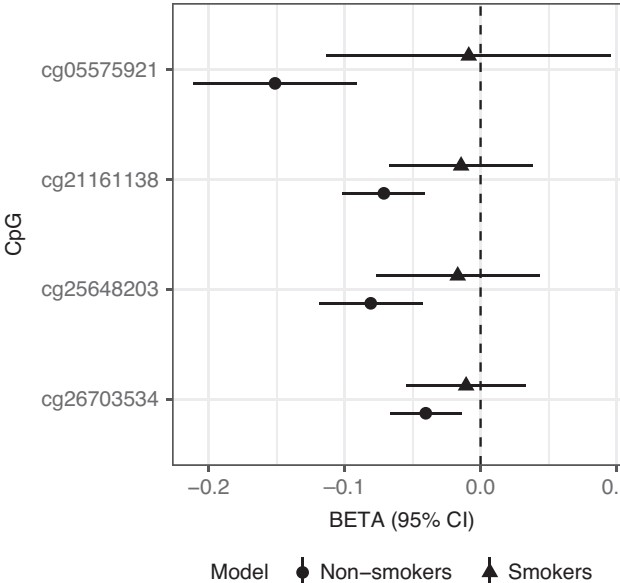

**Fig. 2 Associations between PTSD-associated CpGs stratified by smoking status.** One-thousand eight-hundred and forty-four samples are examined, including 1255 non-smokers (circles) and 589 smokers (triangles). On the x-axis, error bars represent the 95% confidence interval, and the center of the error bars represent effect sizes for each of the CpGs on the y-axis.

the association of *AHRR* and PTSD was no longer significant after statistically controlling for smoking. Marzi and colleagues concluded that there are no robust changes in DNA methylation in victimized young people. Our study went beyond this approach to conduct a stratified analysis that revealed the most prominent PTSD-associated difference in *AHRR* methylation was evident in the non-smokers[24]. To evaluate the possibility of confounding due to unreported smoking among those that identify as non-smokers, we tested top smoking-related CpGs, including *AHRR*, for association with PTSD, and identified no evidence of association between smoking-related CpG sites and PTSD, supporting our conclusion that the association between *AHRR* CpG sites and PTSD was independent of smoking.

Though it is possible that stratifying on smoking could introduce a collider bias that distorts the stratified analysis because of a SNP or other unmeasured confounder, this scenario seems unlikely given that the associations between *AHRR* CpGs and PTSD are stronger in non-smokers. As such, this cross-sectional analysis is unable to determine whether methylation of *AHRR* CpGs is likely a cause or consequence of smoking. However, two recently published studies have provided insight into this question. The first sought to identify DNA methylation differences associated with PTSD in blood of 378 PTSD cases and 135 controls from the Translational Research Center for Traumatic Brain Injury and Stress Disorders (TRACTS) cohort[25]. This study used a methylation-based score for smoking as a covariate, and still reported a significant association between PTSD and cg05575921 (*AHRR*). The second examined DNA methylation differences in blood associated with polycyclic aromatic hydrocarbon (PAH) levels unrelated to smoking[26]. In 708 non-smokers from Taiwan, the authors report associations between $PM_{2.5}$ levels, an indicator of regional air quality, and lower methylation at cg05575921. Taken together, these studies suggest that methylation of this CpG can occur through processes other than smoking.

Over the last decade, multiple studies described the role of the aryl-hydrocarbon receptor in the immune system, with specific roles in T cells, B cells, monocytes and dendritic cells (DC)[21,27–29]. The aryl-hydrocarbon receptor (AhR) pathway works in a regulatory capacity. Briefly, when a ligand binds to the AhR, it translocates to the nucleus where it drives expression of its target genes, including the aryl-hydrocarbon receptor repressor (AHRR), which begins a feedback loop in which it can competitively bind the AhR. The AhR pathway can either limit or stimulate an inflammatory response. One mechanism by which this occurs is by promoting differentiation of T cells into T regulatory cells (Tregs) or T helper 17 (Th17), though this appears to be done in a ligand-specific manner. For example, endogenous ligands, such as kynurenine or dietary indoles[30], tend to promote differentiation into Tregs, which reduce the immune response in a self-limiting manner, resulting in expression of anti-inflammatory genes, reduced inflammation, and less Treg generation. In contrast, exogenous ligands, such as dioxin or polyaromatic hydrocarbons in cigarette smoke, tend to promote differentiation into Th17, expression of pro-inflammatory cytokines, heightened inflammation, and activation the drug metabolizing enzymes[27,31,32].

In a subset of subjects with tryptophan metabolite data, we reported that kynurenine levels were significantly lower in subjects with PTSD relative to trauma-exposed controls and that lower methylation of *AHRR* CpGs associated with lower kynurenine and its metabolites. This pattern is similar to what is observed following chronic exposure to nicotine and possibly other AHR-stimulating ligands[33]. This chronic exposure scenario is reflected in our cohort among both controls and PTSD cases by the finding that higher levels of cotinine were associated with decreased levels of kynurenine. Independent of its source, lower levels of kynurenine would likely result in a counter-regulatory increase in pro-inflammatory activity[34], and may help to account for the frequent observation of heightened inflammatory activity observed in subjects with PTSD[35]. Our secondary analysis of DNA methylation and kynurenine was limited to the only cohort with cotinine and metabolomics data available on the subjects included in the methylation study, which limited sample size, and power for this analysis, and potentially generalizability to other cohorts. However, we believe that the ability to verify self-reported smoking status with a biological measure and to evaluate our kynurenine hypothesis was a considerable strength of the study. We hope these results will prompt other investigators to replicate and extend these findings.

While smokers are more common among our PTSD group, approximately half of the cases in our cohorts are non-smokers, who exhibited the most prominent associations between *AHRR* CpGs and PTSD. Methylation of *AHRR* CpGs decreases only so much even among heavy smokers, suggesting that there may be limited variability in these subjects and a limited degree to which methylation at this site may be decreased. Though the vast majority of the *AHRR* methylation literature is focused on characterizing its variation in the context of smoking, the types of polycyclic aromatic hydrocarbons that stimulate AHR are present from multiple exogenous sources, including burning wood or charcoal, auto emissions, industrial exhaust, and urban dust[36]. Similarly, participants in the WTC and military cohorts are likely to have substantial levels of occupational exposure that could result in AHR activation[37–39]. Though we were not able to directly assess exposure to endogenous or exogenous AhR ligands in this study, it is reasonable to hypothesize that smoking is just one of many potential environmental exposures that could promote inflammation following AHR stimulation.

Both epidemiologic and immunologic studies report immune dysregulation in those with PTSD compared to controls. For example, autoimmune and inflammatory disorders, such as rheumatoid arthritis, have been linked to PTSD[40–42]. Studies of the immune system generally support higher levels of

pro-inflammatory cytokines in PTSD cases relative to controls[35]. A study investigating Tregs reported a lower number of Tregs in the blood of PTSD patients[43], while another reported a difference in the composition of Treg populations that suggested higher susceptibility for autoimmunity[44]. Another study reported lower proportions of Tregs and higher proportions of Th17 cells in PTSD cases before linking Th17 counts to higher clinical symptom scores[45]. Finally, a randomized control trial of subjects undergoing narrative exposure psychotherapy noted higher levels of Tregs and lower PTSD symptoms following treatment[46]. Unfortunately, we were unable to directly measure or impute the proportion of different types of T cells in this study. Though the degree to which the immune system is involved remains speculative without functional studies, we hope these data will support more detailed immunophenotyping of subjects with PTSD.

This study has a number of additional limitations that should be considered. First, we used existing data generated on the HumanMethylation450 array, which captures only a fraction of the CpG sites in the genome. Though sequencing methods would capture a larger proportion of the epigenome that may include regions important for PTSD, focusing on this array allowed us to evaluate a larger and more diverse cohort of subjects and enabled processing and analysis of these diverse samples with a common analytic pipeline. Similarly, this meta-analysis uses only cross-sectional data. In order to establish whether these PTSD-associated differences are a cause or consequence of the disorder or both, longitudinal and functional studies will be required. Second, there is phenotypic heterogeneity among the cohorts contributing to this meta-analysis. Some cohorts assess individuals that were recently exposed to trauma while others include individuals with chronic PTSD. Also, few of the cohorts included in this meta-analysis have detailed physical or psychiatric information on subjects prior to trauma exposure, making it difficult to evaluate the role of lifestyle factors, such as obesity, or comorbidities, such as substance use. Thus, it is possible that some of the epigenetic differences observed in this study may have been in place prior to PTSD development, including the possibility of previous episode of PTSD in subjects that no longer meet current diagnostic criteria, or reflect other factors such as differences in genetic background between cases and controls. Third, our study examined whole blood-derived DNA. Though this approach likely captures part of the PTSD sequelae and may be informative for future biomarker studies, it is unlikely to reflect DNA methylation in brain regions most relevant for PTSD. As studies of tissues from PTSD Brain Banks are conducted, it will be important to look for parallels between these different tissues.

Taken together, the results of this study implicate the immune system in PTSD and suggest that epigenetic mechanisms may play a role in that process. A substantial fraction of those diagnosed with PTSD do not respond to pharmacologic or psychological interventions, and clinical and preclinical studies have begun to evaluate strategies to limit inflammation as a first line treatment[47,48]. Future studies should evaluate the role of ligand-specific AHR activation in the development and progression of PTSD.

## Methods

**Post-traumatic stress disorder cohorts and assessments**. The participating cohorts, presented in Table 1, consisted of three civilian cohorts: the Detroit Neighborhood Health Study (DNHS), the Grady Trauma Project (GTP), and the World Trade Center 9/11 First Responders study (WTC); and seven military cohorts: the Army Study to Assess Risk and Resilience in Servicemembers (Army STARRS), the Marine Resiliency Study (MRS), the Injury and Traumatic Stress study (INTRuST), the Prospective Research in Stress-Related Military Operations study (PRISMO), a European and African-American cohort from the Veterans Affairs' Mental Illness Research, Education and Clinical Centers (VA-M), and a

cohort from the National Center for PTSD (VA-NCPTSD). All subjects participating in these studies provided informed consent, and all studies were approved by respective institutional review boards. Current PTSD diagnosis was assessed by each individual cohort in accordance with the harmonization principles adopted by the PGC-PTSD Workgroup[4]. Briefly, diagnosis of current PTSD was based on the diagnostic criteria defined by each cohort's principal investigator (see cohort descriptions in Supplementary Methods for complete details). All control subjects were trauma-exposed, and, if assessed in the respective cohort, control subjects that had a prior history of PTSD were excluded. Covariates were age, sex, genetic ancestry, and smoking status. A total of 1,896 subjects (42% cases) with DNA methylation from whole blood measured using the Illumina HumanMethylation450 BeadChip were selected for inclusion in this meta-analysis.

**Quality control (QC) procedures**. Data contributed to this study were part of 10 different studies, which were each designed to address a study-specific question. In developing the pipeline to analyze this existing data, the PGC-PTSD group issued recommendations for studies to balance potential confounders. All cohorts were instructed on importance of balancing plate layouts, and the potential for residual sources of technical variation was further assessed by examining the association of methylation principal components with chip, position, plate, and demographic characteristics within each study. Each study performed analyses at their site. To ensure consistent QC procedures across all participating cohorts, a set of common scripts were developed and implemented uniformly[49]. ß-values, representative of the proportion of methylation at each probe, were caculated for all CpG sites in each sample. Samples with probe detection call rates <90% and those with an average intensity value of either <50% of the experiment-wide sample mean or <2000 arbitrary units (AU) were excluded. Probes with detection p-values > 0.001 or those based on less than three beads were set to missing as were probes that cross-hybridized between autosomes and sex chromosomes[50]. CpG sites with missing data for >10% of samples within cohorts were excluded from analysis. Normalization of probe distribution was conducted using Beta Mixture Quantile Normalization (BMIQ)[50] after background correction. Density plots of Type I and Type II probes before and after normalization were examined to confirm probe distributions between types were similar. ComBat was used to account for sources of technical variation including batch and positional effects, while preserving variation attributable to study-specific outcomes and covariates that would be used in downstream analyses (e.g., case status or sex). Proportions of CD8, CD4, NK, B cells, monocytes, and granulocytes were estimated using each individual's DNA methylation data based on the approach described by Houseman and colleagues and publicly available reference data (GSE36069)[51,52].

**Statistical analysis**. Within each cohort, logit transformed ß-values (M-values) were modeled by linear regression as a function of PTSD, adjusting for sex (if applicable), age, CD8, CD4, NK, B cell, and monocyte cell proportions, and ancestry using principal components (PCs). For cohorts with available GWAS data (Army STARRS, GTP, INTRuST, MRS, VA-M, VA-NCPTSD), PCs 1-3 were included as covariates. For cohorts without GWAS data (DNHS, PRISMO, WTC), the method described by Barfield and colleagues was used to generate ancestry PCs directly from the methylation data using CpG sites from the HumanMethylation450 array that are located within 0–100 bp of 1000 Genomes Project variants with minor allele frequency > 0.01, and PCs 2–4, which were the components that correlate most with ancestry as recommended by the authors[53]. Using the empirical Bayes method in the R package limma[54], moderated t-statistics were calculated for each CpG site, converted first into one-sided p-values then converted into z-scores to account for direction of effects. Meta-analysis across cohorts was performed by weighting each cohorts' z-scores by its sample size relative to the total meta-analysis sample, summing weighted z-scores across cohorts from which two-sided p-values were calculated. We used the genome-wide significance threshold suggested for HumanMethylation450 BeadChip to adjust for multiple testing ($p < 3.6E-8$)[55]. The iMETHYL database (http://imethyl.iwate-megabank.org/) was used to evaluate whether methylation of AHRR CpGs associated with AHRR expression in PBMCs, the available tissue that most closely resembles whole blood[56].

**Metabolite analysis in the Marine Resiliency Study (MRS)**. Targeted, broad-spectrum metabolomics was performed as previously described[57,58] with minor modifications. Lithium heparin plasma samples were collected and stored at −80 °C until used for analyses in the MRS metabolomics study. Samples were analyzed from 116 participants exposed to military combat; 53 diagnosed with PTSD, and 63 controls without. Ninety microliters of plasma and 10 μl of added stable isotope internal standards were combined, extracted, and analyzed by hydrophilic interaction chromatography, electrospray ionization, and tandem mass spectrometry on a SCIEX 5500 QTRAP HILIC-ESI-MS/MS platform. From the 486 metabolites measured, we extracted and analyzed data for kynurenine and kynurenic acid to test our hypothesis that AHRR methylation associates with tryophan breakdown products. Cotinine, a metabolite of nicotine with a plasma half-life of about 18 h, was measured to provide independent information about tobacco exposure. A cotinine arbitrary unit (AU) of $1 \times 10^6$ was used as the upper

limit for a non-smoker, light smoker, or second-hand tobacco exposure, with $\geq 1 \times 10^6$ indicative of regular or heavy smoking. Analyte AU data was log-transformed, and analyte z-scores were used for further statistical analysis. To determine if trytophan metabolite levels differed between PTSD cases and controls, we performed linear regressions of metabolites on PTSD status, including GWAS PCs 1–3 and cell type proportions as covariates. To determine if tryptophan metabolite levels were associated with AHRR methylation, we performed linear regressions of metabolites on each AHRR CpG site, including PCs and cell type proportions as covariates. Correlation between cotinine and kynurenine was measured using Pearson's correlation coefficient.

**Reporting summary**. Further information on research design is available in the Nature Research Reporting Summary linked to this article.

## Data availability
The main summary statistics data that support the findings of this study are available within Supplementary Data 2. Individual-level data from the cohorts or cohort-level summary statistics will be made available to researchers following an approved analysis proposal through the PGC Post-traumatic Stress Disorder group with agreement of the cohort PIs. The raw data for the GTP cohort is available in the Gene Expression Omnibus database with the accession code GSE72680. Owing to military cohort data sharing restrictions, data from the VA, VA MIRECC, MRS, Army STARRS, and PRISMO cannot be publicly posted. However, such data can be provided in de-identified from a data repository through a data use agreement following applicable guidelines on data sharing and privacy protection. For additional information on access to these data, including PI contact information for the contributing cohorts, please contact the corresponding author.

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

## Acknowledgements

This work was supported by the U.S. Army Medical Research and Materiel Command and the National Institute of Mental Health (NIMH; R01MH108826; R01MH106595), as well as the Biomedical and Laboratory Research and Development (#I01BX002577). We appreciate the technical support of all of the staff, volunteers and participants from the Grady Trauma Project, supported by the National Institutes of Mental Health (MH096764 and MH071537). DNHS, which is grateful to all of the participants and staff for their contributions, was funded by NIH Awards R01DA022720, R01DA022720-S1, and RC1MH088283. The Marine Corps, Navy Bureau of Medicine and Surgery (BUMED) and VA Health Research and Development (HSR&D) provided funding for MRS data collection and analysis and NIH R01MH093500 funded the GWAS assays and analysis. Acknowledged are Mark A. Geyer (UCSD), Daniel T. O'Connor (UCSD), all MRS investigators, as well as the MRS administrative core and data collection staff listed in the Methods article[59]. We also thank the Marine and Navy Corpsmen volunteers for military service and participation in MRS. Support for metabolomics in the Naviaux lab at UCSD was provided in part by philanthropic gifts from the UCSD Christini Fund, the Lennox Foundation, the Wright Family Foundation, and the Jane Botsford Johnson Foundation. Data collection of PRISMO was funded by the Dutch Ministry of Defense, and DNA methylation analyses were funded by the VENI Award fellowship from the Netherlands Organization for Scientific Research (NWO, grant number 916.11.086). The VA Boston-National Center for PTSD Study research was supported in part by National Institute of Mental Health Award RO1MH079806, Department of Veterans Affairs, Clinical Science Research & Development Program Award 5I01CX000431-02, Department of Veterans Affairs, Biomedical Laboratory Research & Development Program Award 1I01BX002150-01. This research is the result of work supported with resources and the use of facilities at the Pharmacogenomics Analysis Laboratory, Research and Development Service, Central Arkansas Veterans Healthcare System, Little Rock, Arkansas. This work was also supported by a Career Development Award to E. J. Wolf from the Department of Veterans Affairs, Clinical Sciences Research, and Development Program. Dr. Kimbrel was supported by a Career Development Award (#IK2CX000525) from the Clinical Science Research and Development (CSR&D). Dr. Beckham was supported by a Research Career Scientist Award (#11S-RCS-009) from the CSR&D Service of VA ORD. This research was also supported, in part, by a Merit Award (#I01BX002577) from the Biomedical Laboratory Research and Development (BLR&D) Service of VA ORD. The VA Mid-Atlantic Mental Illness Research, Education, and Clinical Center Workgroup includes John A. Fairbank, Mira Brancu, Patrick S. Calhoun, Eric A. Dedert, Eric B. Elbogen, Kimberly T. Green, Robin A. Hurley, Angela C. Kirby, Jason D. Kilts, Christine E. Marx, Gregory McCarthy, Scott D. McDonald, Marinell Miller-Mumford, Scott D. Moore, Rajendra A. Morey, Jennifer C. Naylor, Treven C. Pickett, Jared Rowland, Jennifer J. Runnals, Cindy Swinkels, Steven T. Szabo, Katherine H. Taber, Larry A. Tupler, Elizabeth E. Van Voorhees, H. Ryan Wagner, Richard D. Weiner, and Ruth Yoash-Gantz. Army STARRS was sponsored by the Department of the Army and funded under cooperative agreement number U01MH087981 (2009-2015) with the National Institutes of Health, National Institute of Mental Health (NIH/NIMH). WTC study was sponsored by CDC/NIOSH award U01 OH010416-01. The PTSD and TBI INjury and TRaUmatic STress Clinical Consortium (INTRuST) was funded by a grant from the United States Department of Defense: W81XWH08-2-0159. Members of the INTRuST Consortium Biorepository Working Group who contributed to this work include: Gerald A. Grant MD, Christine E. Marx MD, Mark S. George MD, Thomas W. McAllister MD, Norberto Andaluz MD, Lori Shutter MD, Raul Comibra MD, Ross D. Zafonte DO, Sonia Jain PhD, Xue-Jun Qin, and Michael Hauser PhD. The views expressed in this article are those of the authors and do not necessarily reflect the position or policy of the VA, NIH, or the United States government.

## Author contributions

Interpreted results, writing, and editing the manuscript: A.K.S., A.R., A.X.M., R.K.N., M.W.L., M.U., C.M.N. Conceptualization and supervision of project: A.K.S., M.W.L., M.U., C.M.N. Performed meta-analyses: A.R., A.X.M., S.K. Performed metabolomics analyses: A.X.M., R.K.N., K.L., J.C.N. Contributed to study-specific EWAS by providing phenotype, genotype and performing data analyses: A.K.S., A.R., A.X.M., A.E.A., A.B.A., A.E.A.K., D.G.B., J.C.B., M.P.B., E.B., M.D., S.G., M.E.G., E.G., G.G., M.A.H., S.K., V.K., R.C.K., N.A.K., K.C.K., P.F.K., M.W.L., A.L., B.J.L., M.W.M., N.R.N., X.Q., K.J.R., V.B.R., B.P.F.R., M.B.S., R.J.U., E.V., C.H.V., L.W., N.A.Y., M.U., C.M.N. Unless otherwise stated, INTRuST Clinical Consortium, V.A. Mid-Atlantic MIRECC Workgroup and PGC-PTSD Epigenetics Workgroup authors contributed to data collection but did not participate in analysis or writing of this report. All other authors reviewed and approved the final version of the paper.

## Competing interests

Dr. Youssef's disclosures include Speaker CME honoraria from the Georgia Department of Behavioral Health and Developmental Disabilities (DHBDD). Dr. Youssef received research support from the Department of Veteran Affairs and The Augusta Biomedical Research Corporation in the last 3 years. His current research funding (but not direct payment) include Merck pharmaceuticals (8S0073-9), and MECTA Corporation (72200S-2) and American Foundation for Suicide Prevention. Dr. Stein has in the past 3 years received payments for editorial work from UpToDate, Biological Psychiatry, and Depression and Anxiety. He has also in the past 3 years been paid as a consultant for Actelion Pharmaceuticals, Aptinyx, Bionomics, Janssen, and Pfizer. No other author declares any conflict of interest.

## Additional information

Alicia K. Smith [1,2✉], Andrew Ratanatharathorn [3], Adam X. Maihofer [4], Robert K. Naviaux[5], Allison E. Aiello[6], Ananda B. Amstadter[7], Allison E. Ashley-Koch [8], Dewleen G. Baker [4,9,10], Jean C. Beckham[11,12,13], Marco P. Boks[14], Evelyn Bromet[15], Michelle Dennis[11,13], Sandro Galea [16], Melanie E. Garrett[8], Elbert Geuze [14,17], Guia Guffanti[18,19], Michael A. Hauser[8,20], Seyma Katrinli [1], Varun Kilaru[1], Ronald C. Kessler [19], Nathan A. Kimbrel[11,12,13], Karestan C. Koenen[21,22,23], Pei-Fen Kuan [24], Kefeng Li [25], Mark W. Logue[26,27,28,29], Adriana Lori[2], Benjamin J. Luft [30], Mark W. Miller [26,27], Jane C. Naviaux[31], Nicole R. Nugent[32], Xuejun Qin [8], Kerry J. Ressler [2,18,19], Victoria B. Risbrough [4,9,10], Bart P. F. Rutten [33], Murray B. Stein [4,9,34], Robert J. Ursano [35], Eric Vermetten [14,35,36,37,38], Christiaan H. Vinkers [39,40], Lin Wang[25], Nagy A. Youssef[41], INTRuST Clinical Consortium*, VA Mid-Atlantic MIRECC Workgroup*, PGC PTSD Epigenetics Workgroup*, Monica Uddin[42] & Caroline M. Nievergelt[4,9,10]

[1]Emory University, Department of Gynecology and Obstetrics, Atlanta, GA, USA. [2]Emory University, Department of Psychiatry & Behavioral Sciences, Atlanta, GA, USA. [3]Columbia University, Department of Epidemiology, New York, NY, USA. [4]University of California San Diego, Department of Psychiatry, La Jolla, CA, USA. [5]University of California, The Mitochondrial and Metabolic Disease Center, Departments of Medicine, Pediatrics, and Pathology, San Diego, CA, USA. [6]University of North Carolina, Gillings School of Global Public Health, Department of Epidemiology, Chapel Hill, NC, USA. [7]Virginia Commonwealth University, Department of Psychiatry, Richmond, VA, USA. [8]Duke Molecular Physiology Institute, Duke University Medical Center, Durham, NC, USA. [9]Veterans Affairs San Diego Healthcare System, San Diego, CA, USA. [10]Veterans Affairs Center of Excellence for Stress and Mental Health, San Diego, CA, USA. [11]VA Mid-Atlantic, Mental Illness Research, Education, and Clinical Center, Durham, NC, USA. [12]Veterans Affairs Durham Healthcare System, Durham, NC, USA. [13]Duke University Medical Center, Department of Psychiatry and Behavioral Sciences, Durham, NC, USA. [14]University Medical Center Utrecht, Brain Center Rudolf Magnus, Utrecht, The Netherlands. [15]State University of New York at Stony Brook, Epidemiology Research Group, Stony Brook, NY, USA. [16]Boston University, School of Public Health, Boston, MA, USA. [17]Netherlands Ministry of Defence, Brain Research and Innovation Centre, Utrecht, The Netherlands. [18]McLean Hospital, Division of Depression and Anxiety, Belmont, MA, USA. [19]Harvard Medical School, Boston, MA, USA. [20]Duke University, Department of Medicine, Durham, NC, USA. [21]Harvard T.H. Chan School of Public Health, Department of Epidemiology, Boston, MA, USA. [22]Massachusetts General Hospital, Psychiatric and Neurodevelopmental Genetics Unit, Center for Human Genetic Research, and Department of Psychiatry, Boston, MA, USA. [23]Broad Institute of MIT and Harvard, Stanley Center for Psychiatric Research, Cambridge, MA, USA. [24]State University of New York at Stony Brook, Department of Applied Mathematics and Statistics, Stony Brook, NY, USA. [25]University of California, The Mitochondrial and Metabolic Disease Center, Department of Medicine, San Diego, CA, USA. [26]National Center for PTSD, Behavioral Science Division at VA Boston Healthcare System, Boston, MA, USA. [27]Boston University School of Medicine, Department of Psychiatry, Boston, MA, USA. [28]Boston University School of Medicine, Department of Medicine (Biomedical Genetics), Boston, MA, USA. [29]Boston University School of Public Health, Department of Biostatistics, Boston, MA, USA. [30]State University of New York at Stony Brook, Department of Medicine, Stony Brook, NY, USA. [31]University of California, The Mitochondrial and Metabolic Disease Center, Department of Neuroscience, San Diego, CA, USA. [32]Brown University, Psychiatry and Human Behavior, Department of Pediatric Research, Providence, RI, USA. [33]Maastricht University Medical Centre, School for Mental Health and Neuroscience, Department of Psychiatry and Neuropsychology, Maastricht, The Netherlands. [34]University of California San Diego, Department of Family Medicine and Public Health, La Jolla, CA, USA. [35]Uniformed Services University School of Medicine, Center for the Study of Traumatic Stress, Bethesda, MD, USA. [36]Leiden University Medical Center, Department of Psychiatry, Leiden, The Netherlands. [37]Netherlands Defense Department, Research Center, Utrecht, UT, The Netherlands. [38]Arq Psychotrauma Expert Group, Diemen, The Netherlands. [39]Amsterdam UMC (location VUmc), Department of Psychiatry, Amsterdam, The Netherlands. [40]Amsterdam UMC (location VUmc), Department of Anatomy and Neurosciences, Amsterdam, The Netherlands. [41]Medical College of Georgia at Augusta University, Department of Psychiatry and Human Behavior and Office of Academic Affairs, Augusta, GA, USA. [42]University of South Florida, College of Public Health, Tampa, FL, USA. *Lists of authors and their affiliations appear at the end of the paper. ✉email: alicia.smith@emory.edu

## INTRuST Clinical Consortium

Christine Marx[11,12,13], Gerry Grant[43], Murray Stein [4,9,34], Michael A. Hauser[8,20], Xue-Jun Qin[8], Sonia Jain[34], Thomas W. McAllister[44], Ross Zafonte[45], Ariel Lang[4], Raul Coimbra[46], Norberto Andaluz[47], Lori Shutter[48] & Mark S. George[49]

[43]Department of Neurosurgery, Stanford University Medical Center, Stanford, CA, USA. [44]Department of Psychiatry, Indiana University, Indianapolis, IN, USA. [45]Spaulding Rehabilitation Hospital and Harvard Medical School, Charleston, MA, USA. [46]Riverside University Health System Medical Center, Moreno Valley, CA, USA. [47]University of Louisville School of Medicine, Louisville, KY, USA. [48]University of Pittsburgh School of Medicine, Pittsburgh, PA, USA. [49]Medical University of South Carolina, Ralph H. Johnson VA Medical Center, Charleston, SC, USA.

## VA Mid-Atlantic MIRECC Workgroup

Mira Brancu[12,20], Patrick S. Calhoun[12,20], Eric Dedert[12,20], Eric B. Elbogen[12,20], John A. Fairbank[12,20], Robin A. Hurley[50,51,52], Jason D. Kilts[12,20], Angela Kirby[12], Christine E. Marx[12,20], Scott D. McDonald[7,53], Scott D. Moore[12,20], Rajendra A. Morey[12,20], Jennifer C. Naylor[12,20], Jared A. Rowland[50,51], Cindy Swinkels[12,20],

Steven T. Szabo[12,20], Katherine H. Taber[50,52,54], Larry A. Tupler[12,20], Elizabeth E. Van Voorhees[12,20] & Ruth E. Yoash-Gantz[50,51]

[50]Salisbury Veterans Affairs Health Care System, Salisbury, NC, USA. [51]Wake Forest University, Winston-Salem, NC, USA. [52]Baylor College of Medicine, Houston, TX, USA. [53]Richmond Veterans Affairs Health Care System, Richmond, VA, USA. [54]Virginia College of Osteopathic Medicine, Blacksburg, VA, USA.

## PGC PTSD Epigenetics Workgroup

Allison E. Aiello[6], Ananda B. Amstadter[7], Allison E. Ashley-Koch[8], Dewleen G. Baker[4,9,10], Archana Basu[21], Jean C. Beckham[11,12,13], Marco P. Boks[14], Leslie A. Brick[32], Evelyn Bromet[15], Shareefa Dalvie[55], Nikolaos P. Daskalakis[19,56,57], Michelle Dennis[11,13], Judith B. M. Ensink[58,59,60], Sandro Galea[16], Melanie E. Garrett[8], Elbert Geuze[14,17], Guia Guffanti[18,19], Michael A. Hauser[8,20], Sian M. J. Hemmings[61], Ryan Herringa[62], Sylvanus Ikiyo[63], Seyma Katrinli[1], Ronald C. Kessler[19], Varun Kilaru[1], Nathan A. Kimbrel[11,12,13], Nastassja Koen[55], Karestan C. Koenen[21,22,23], Pei Fen Kuan[24], Mark W. Logue[26,27,28,29], Adriana Lori[2], Benjamin J. Luft[30], Adam X. Maihofer[4], Mark W. Miller[26,27], Janitza Montalvo-Ortiz[64,65], Caroline M. Nievergelt[4,9,10], Danny Nispeling[14], Nicole R. Nugent[32], John Pfeiffer[66], XueJun Qin[8], Andrew Ratanatharathorn[3], Kerry J. Ressler[2,8,19], Victoria B. Risbrough[4,9,10], Bart P. F. Rutten[33], Dick Schijven[14], Soraya Seedat[61], Gen Shinozaki[67], Alicia K. Smith[1,2✉], Murray B. Stein[4,9,34], Jennifer A. Sumner[68], Patricia Swart[69], Audrey Tyrka[32,70], Monica Uddin[42], Robert J. Ursano[35], Mirjam Van Zuiden[39], Eric Vermetten[14,35,36,37,38], Christiaan H. Vinkers[39,40] & Agaz Wani[42], Erika J. Wolf[26,27], Nagy A. Youssef[41] & Anthony Zannas[13,71,72,73,74]

[55]University of Cape Town, SA MRC Unit on Risk & Resilience in Mental Disorders, Department of Psychiatry, Cape Town, ZA, South Africa. [56]Cohen Veterans Bioscience, Cambridge, USA. [57]Icahn School of Medicine at Mount Sinai, Department of Psychiatry, New York, NY, USA. [58]University of Amsterdam, Department of Child and Adolescent Psychiatry, Amsterdam Public Health Research Institute, Amsterdam UMC, Location Academic Medical Center, Amsterdam, The Netherlands. [59]Academic Center for Child and Adolescent Psychiatry, De Bascule, Amsterdam, The Netherlands. [60]University of Amsterdam, Department of Clinical Epidemiology, Biostatistics and Bioinformatics, Amsterdam Public Health Research Institute, Amsterdam UMC, Location Academic Medical Center, Amsterdam, The Netherlands. [61]Stellenbosch University Faculty of Medicine and Health Sciences, Department of Psychiatry, Cape Town, ZA, South Africa. [62]University of Wisconsin School of Medicine and Public Health, Department of Psychiatry, Madison, WI, USA. [63]University of Aberdeen, Department of Applied Medicine, Aberdeen, UK. [64]Yale University School of Medicine, Division of Human Genetics, Department of Psychiatry, New Haven, CT, USA. [65]VA CT Healthcare Center, West Haven, CT, USA. [66]University of Illinois at Urbana-Champaign, Department of Psychology, Urbana, IL, USA. [67]University of Iowa Carver College of Medicine, University of Iowa Hospitals and Clinics, Department of Psychiatry, Iowa City, IA, USA. [68]Columbia University Medical Center, Department of Medicine, New York, NY, USA. [69]University of Cape Town, Department of Human Biology, Cape Town, ZA, South Africa. [70]Butler Hospital, Mood Disorders Research Program and Laboratory for Clinical and Translational Neuroscience, Providence, RI, USA. [71]University of North Carolina, Department of Psychiatry, Chapel Hill, NC, USA. [72]University of North Carolina, Department of Genetics, Chapel Hill, NC, USA. [73]University of North Carolina School of Medicine, Institute for Trauma Recovery, Chapel Hill, NC, USA. [74]University of North Carolina, Neuroscience Curriculum, Chapel Hill, NC, USA.

