## [Peer Review File · Nature Communications]

Reviewers' comments:

Reviewer #1 (Remarks to the Author):

This is an interesting meta-analysis project, which explores the association between PTSD and DNA methylation from blood. The data and hypotheses are interesting, although I have a few major comments/suggestions:

Line 147 (Table 1): There are lots of potential strata within the study cohorts e.g., sex, smoking, ethnicity. Only sex is controlled for in the EWAS model, though smoking is discussed in detail in secondary analyses. Were there consistent effect size estimates seen across cohorts and ethnicities? It would be good to see forest plots for the various strata.

Line 158: Please quantify what is meant by moderately correlated. I have had a look in one of the datasets I have access to and main AHRR probe (cg05575921) is correlated between 0.48 and 0.59 with the other three probes.

Line 156: Why not use the genome wide threshold for EWAS significance ($P < 3.6 \times 10^{-8}$ - <https://www.ncbi.nlm.nih.gov/pubmed/29034560>)? Using FDR without any form of replication does not seem robust enough to me.

Line 176: I think a more convincing way to remove former smokers and un-reported smoking in non-smokers would be to generate an epigenetic score for smoking (e.g., <https://genomebiology.biomedcentral.com/articles/10.1186/s13059-018-1514-1>) and to include that as a covariate in the non-smoker analysis.

Line 196-198: Given that the AHRR is so strongly associated with smoking status, I am not sure that we'd expect to see similar associations across other smoking-associated CpGs. The AHRR probes on their own explain nearly all of the difference between current/ex/never smokers.

Minor comments:

Line 115: I'm not sure " β " will be completely obvious to a non-EWAS expert.

Line 118: Similarly, unless you are an EWAS/Epigenetic Clock expert, the concept of age acceleration will not be immediately obvious. I think a slightly more detailed explanation would be helpful here.

Line 121: "low statistical power" I think this could be deleted or qualified. For example, low power to detect what? Standardised effect sizes greater than x % difference between cases and controls?

Line 131: 42% with a current diagnosis but what about former diagnoses?

Line 186-190: Please could you re-phrase/re-write this section? I found it tricky to follow.

Line 216: I think empiric should be empirical

Line 217: What does an AUC $< 1 \times 10^{-6}$ mean? I am used to AUC being presented on a scale between 0.5 and 1.

Line 239: Did the cohorts used in the current study contribute to these prior analyses?

Reviewer #2 (Remarks to the Author):

In this interesting meta-analysis, the authors leveraged 10 military and civilian cohorts to examine the association of PTSD diagnosis with DNA (CpG) methylation at the epigenome-wide level. The manuscript is well-written and the study has several strengths: a) it is the largest meta-analysis to date associating PTSD and DNA methylation ($n > 1,800$); b) it examines both military and civilian cohorts with harmonized procedures according to the PGC PTSD Epigenetics workgroup; c) it uses state-of-the-art methods to control for confounding by blood cell proportions and smoking; and d) it includes metabolomic analyses, though in a single cohort and much smaller sample ($n=116$), which nicely complement the DNA methylation findings. Overall the study identifies 10 novel CpGs associated with PTSD. Among these, most interesting is the association of PTSD with lower methylation at 4 CpGs (which are also the overall top associated sites) located in the gene body of AHRR, spanning as the authors point out a region of about 22kb. Further, the authors find associations with kynurenin levels and attempt to link this finding with immune dysregulation. Points to be addressed include the following:

- 1) The abstract occasionally lacks sufficient clarity. In line 87, "Several were located in genes implicated in psychiatric disorders", does 'several' refer to the 10 CpGs or to the 50 CpGs mentioned later in the results section? In line 89, the phrase "appeared to uncorrelated" needs to be corrected. The abstract also does not mention to readers why kynurenine levels were examined and why this is relevant for immune dysregulation.
- 2) In line 158, the authors describe "though methylation of these CpGs are moderately correlated," but they do not provide statistics for individual CpGs, which would help readers better contextualize this finding.
- 3) Could the observed associations be confounded by other lifestyle parameters, e.g. diet, which could be controlled for using metrics such as BMI, or other substance use, such as cannabis or secondarily cocaine and opioids? If the authors have such data in at least some of the participating cohorts, it would be worth controlling for in their analyses.
- 4) On several occasions (e.g. all analyses including metabolomic data), the authors provide only p-values. Detailed stats should be consistently reported throughout the manuscript to give a better sense of effect size.
- 5) Could the authors provide a brief explanation as to why PCs 2-4 were used to adjust for ancestry (line 411)? Not having to refer to the cited article would help readers.
- 6) The authors provide overall sufficient explanation concerning the pipeline used to control for technical batches (lines 396-399). What is missing, however, is more detail on how samples were randomized across plates, slides, rows, and columns of the HumanMethylation450 BeadChip arrays. In particular, did the authors confirm balanced distribution of the critical variables, i.e. PTSD case control status, age, sex, and smoking? Such a balanced distribution is a prerequisite to allow for sufficient adjustment for batch effects.
- 7) In figure S1, what is the reason for the larger box for PRISMO as compared to the other cohorts? Does box size reflect weight, and was this mainly based on sample size? This should be clarified, especially since PRISMO is the cohort with the smallest sample size (at least according to table 1).
- 8) In the same figure S1 (or in a separate figure), it would be helpful to present the confidence intervals stratified (for both smokers and nonsmokers) in each cohort.
- 9) The authors readily conclude that lower AHRR gene body methylation also means higher expression of AHRR; however, they have no functional data to support such a claim, which is not necessarily true given that gene body CpG methylation has been shown to positively correlate with gene expression (for example, see: Ball et al., Nat Biotechnol 2009;27(4):361-8, doi: 10.1038/nbt.1533). This becomes problematic because the whole manuscript is then built on this ungrounded assumption.
- 10) Other aspects of the overall model (as presented on Figure 3) are also too speculative. Why do the authors assume that PTSD cases had more exposure to AhR ligands? Could they provide relevant data to support this claim or at least previous evidence from human cohorts showing regulation of the pathway depending on AhR ligand exposure? Could changes in methylation simply reflect other factors, e.g. differences in genetic background between cases and controls? Likewise, the authors

have no data on Th17 or Treg to support differences in immune cell population and function.

Reviewer #3 (Remarks to the Author):

The authors report associations of several CpG sites with post-traumatic stress disorder from a meta-analysis of ten cohorts. The top hits are in credible genes with results of particular interest in AHRR. As the methylation of this gene is associated with smoking, and there are more smokers among PTSD cases within cohorts, the authors performed adjusted and stratified analyses for smoking. Although the associations were attenuated, they remained significant among non-smokers suggesting a direct association in addition to possible confounding. I have a few minor comments.

1. Figure 1 shows significance thresholds for two FDR levels. This doesn't make sense as the FDR adjustment is different for every CpG depending on its ranked p-value.
2. P8 L181 "collinearity" isn't really the right term as there is still a good proportion of non-smokers among the cases. I'd suggest that what you have shown is more like an interaction between PTSD and smoking status on AHRR methylation. If one is a smoker there isn't much of an effect of PTSD, whereas if one is a non-smoker then PTSD status has more of an effect.
3. P10 L217-219 I didn't understand the meaning of "AUC < 1e6" and "AUC >= 1e7). If this is area under the ROC curve then it should be between 0.5 and 1.
4. Is it assumed that AHRR demethylation is a consequence of smoking - presumably so, would be helpful to clarify. Did you consider whether smoking is a cause or consequence of PTSD? This doesn't have much bearing on the results, except if smoking is a consequence of PTSD and there are unmeasured confounders of the smoking-AHRR association. Then, conditioning/stratifying on smoking could introduce a collider bias that distorts the stratified analysis (such a confounder might be a SNP). I suspect this scenario is somewhat unlikely, especially as the association is stronger in non-smokers, but it might be worth a comment.
5. Figure S2, some of the p-values are given as <0.001 where the CIs appear to touch or overlap the null - doesn't look right.

NCOMMS-19-18911

We appreciate the reviewer's comments on the potential importance of the study as well as their suggestions to improve our analysis. Based on their comments and the editor's suggestions, we have clarified methods, performed stratified analyses, implemented a more stringent significance threshold, and provided additional support for the assumption that *AHRR* methylation associates with expression levels. Despite this extensive revision, the main conclusions of the paper are unchanged and, we believe, better supported. Below is a point-by-point response to each concern raised during the initial review.

Reviewer 1:

Line 147 (Table 1): There are lots of potential strata within the study cohorts e.g., sex, smoking, ethnicity. Only sex is controlled for in the EWAS model, though smoking is discussed in detail in secondary analyses. Were there consistent effect size estimates seen across cohorts and ethnicities? It would be good to see forest plots for the various strata.

In the revised submission, we have more completely explored the potential strata within the dataset and their influence on the results. In the supplemental data (Figures S2, S3, and S5), we now provide plots stratified by sex, ancestry, and smoking status. We performed ancestry-specific analyses and meta analyzed the results. The effect sizes for the top hits were consistent across different strata, with significant associations between PTSD and the top CpGs in the following strata: males, females, European ancestry, African ancestry, and nonsmokers. The only strata that did not show this association was smokers, which is consistent with our initial findings.

Line 158: Please quantify what is meant by moderately correlated. I have had a look in one of the datasets I have access to and main *AHRR* probe (cg05575921) is correlated between 0.48 and 0.59 with the other three probes.

We agree that the term “moderately correlated” is vague and subjective. Therefore, we removed that phrase from the text and included correlation coefficients instead.

Line 156: Why not use the genome wide threshold for EWAS significance ($P < 3.6 \times 10^{-8}$ - <https://www.ncbi.nlm.nih.gov/pubmed/29034560>)? Using FDR without any form of replication does not seem robust enough to me.

Thank you for your suggestion. We agree that the genome-wide threshold for EWAS significance ($P < 3.6 \times 10^{-8}$) is more robust and have updated our results and discussion accordingly. Notably, the top 4 *AHRR* probes remain significant.

Line 176: I think a more convincing way to remove former smokers and un-reported smoking in non-smokers would be to generate an epigenetic score for smoking (e.g., <https://genomebiology.biomedcentral.com/articles/10.1186/s13059-018-1514-1>) and to include that as a covariate in the non-smoker analysis.

The CpGs that make up the epigenetic smoking score cited is made of up of 233 CpGs and includes CpGs that associate with smoking from the CHARGE consortium analysis (e.g. cg01940273). It also includes cg05575921 (*AHRR*), thus complicating our ability to interpret either association or lack of association for this individual CpG.

However, we would like to draw attention to the results of 2 recently published studies, which are now discussed in the revised manuscript. The first leverages the MethylationEPIC array to identify DNA methylation differences associated with PTSD in blood of 378 PTSD cases and 135 controls from the Translational Research Center for TBI and Stress Disorders (TRACTS) cohort (PMID: 32171335). This study used a methylation-based score for smoking as a covariate, and still reported a significant association between PTSD and cg05575921 (*AHRR*; $p=9.16E-6$). The second examines whether there are DNA methylation differences in blood related to polycyclic aromatic hydrocarbon (PAH) levels that were unrelated to smoking (PMID: 31060609). In 708 non-smokers from Taiwan, the authors report associations between $PM_{2.5}$ levels, an indicator of regional air quality, and lower methylation at cg05575921. Taken together, these data suggest that methylation of this CpG can occur through processes other than smoking.

Line 196-198: Given that the *AHRR* is so strongly associated with smoking status, I am not sure that we'd expect to see similar associations across other smoking-associated CpGs. The *AHRR* probes on their own explain nearly all of the difference between current/ex/never smokers.

The CpGs tested were the most strongly associated with smoking from the CHARGE consortium analysis of 15,907 blood derived DNA samples from participants in 16 cohorts. Though *AHRR* CpGs associate with smoking, they are not among those most significantly associated with smoking. At no point did we control for *AHRR* methylation in our examination of the other smoking-associated CpGs so there is no reason that we should not have been able to detect associations independent of *AHRR* methylation.

We agree that *AHRR* methylation is highly *sensitive* to smoking. However, a key conclusion of this paper, and the two discussed above (PMID: 32171335 and PMID: 31060609), is that that *AHRR* methylation may not be *specific* to smoking as it may be activated though other endogenous or exogenous ligands.

Line 115: I'm not sure " β " will be completely obvious to a non-EWAS expert.

We have clarified the term (β) by defining it is 'the proportion of methylated DNA intensity to non-methylated DNA intensity'.

Line 118: Similarly, unless you are an EWAS/Epigenetic Clock expert, the concept of age acceleration will not be immediately obvious. I think a slightly more detailed explanation would be helpful here.

We have clarified the definition of age acceleration by stating that it is 'an epigenetic biomarker of aging' and included additional citations.

Line 121: "low statistical power" I think this could be deleted or qualified. For example, low power to detect what? Standardised effect sizes greater than x % difference between cases and controls?

We removed "low statistical power" from the text.

Line 131: 42% with a current diagnosis but what about former diagnoses?

Since DNA methylation is responsive to the environment, it likely reflective of current environmental influences. Thus, we focused our analysis on current PTSD status at the time of blood draw. We agree that lifetime history of PTSD is also relevant. Unfortunately, many of the cohorts that participated in this meta-analysis do not capture lifetime data. We now discuss this as a limitation.

Line 186-190: Please could you re-phrase/re-write this section? I found it tricky to follow.

Thank you. We rephrased this section and hope it is now more clear.

Line 216: I think empiric should be empirical

We corrected empiric to empirical.

Line 217: What does an AUC $<1 \times 10^6$ mean? I am used to AUC being presented on a scale between 0.5 and 1.

This was a typographical error for which we apologize. Cotinine is determined based on arbitrary units (AU) and not an area under the curve. This has been corrected in the manuscript.

Line 239: Did the cohorts used in the current study contribute to these prior analyses?

Only a subset of the MRS cohort was used in a prior analysis (PMID: 25754082).

Reviewer 2:

1) The abstract occasionally lacks sufficient clarity. In line 87, "Several were located in genes implicated in psychiatric disorders", does 'several' refer to the 10 CpGs or to the 50 CpGs mentioned later in the results section? In line 89, the phrase "appeared to uncorrelated" needs to be corrected. The abstract also does not mention to readers why kynurenine levels were examined and why this is relevant for immune dysregulation.

We changed the abstract to clarify and include missing information.

2) In line 158, the authors describe "though methylation of these CpGs are moderately correlated," but they do not provide statistics for individual CpGs, which would help readers better contextualize this finding.

We agree that the term “moderately correlated” is vague and subjective. Therefore, we removed that phrase from the text and included correlation coefficients instead.

3) Could the observed associations be confounded by other lifestyle parameters, e.g. diet, which could be controlled for using metrics such as BMI, or other substance use, such as cannabis or secondarily cocaine and opioids? If the authors have such data in at least some of the participating cohorts, it would be worth controlling for in their analyses.

Unfortunately, the majority of the contributing cohorts do not have data regarding lifestyle parameters including BMI and substance use. We have more thoroughly discussed this as a limitation of the meta-analysis.

4) On several occasions (e.g. all analyses including metabolomic data), the authors provide only p-values. Detailed stats should be consistently reported throughout the manuscript to give a better sense of effect size.

The detailed statistics on metabolite analysis are not included in the supplement.

5) Could the authors provide a brief explanation as to why PCs 2-4 were used to adjust for ancestry (line 411)? Not having to refer to the cited article would help readers.

We revised the text to clarify that PC2-4 are the components that correlate most with ancestry as documented in the initial study (PMID: 24478250). In a subset of PTSD subjects with GWAS data, we have confirmed that PC2-4 accurately capture GWAS-based ancestry PCs. (PMID: 28691784).

6) The authors provide overall sufficient explanation concerning the pipeline used to control for technical batches (lines 396-399). What is missing, however, is more detail on how samples were randomized across plates, slides, rows, and columns of the HumanMethylation450 BeadChip arrays. In particular, did the authors confirm balanced distribution of the critical variables, i.e. PTSD case control status, age, sex, and smoking? Such a balanced distribution is a prerequisite to allow for sufficient adjustment for batch effects.

Data contributed to this study were part of 10 different studies, which were each designed to address a study-specific question. As such, some were cross-sectional and others were longitudinal. In developing the pipeline to analyze this existing data, the PGC PTSD group issued recommendations for studies to balance potential confounders. We have expanded our discussion of these recommendations and the post-hoc evaluation that was used to assess whether residual sources of technical variation remained in the Methods. We can confirm that there was no evidence of bias for PTSD case status, age, sex, or smoking in any contributing study.

7) In figure S1, what is the reason for the larger box for PRISMO as compared to the other cohorts? Does box size reflect weight, and was this mainly based on sample size? This

should be clarified, especially since PRISMO is the cohort with the smallest sample size (at least according to table 1).

Box size was proportional to the standard errors. Note: PRISMO was reanalyzed with a corrected pipeline and its standard errors are now in line with the other studies.

8) In the same figure S1 (or in a separate figure), it would be helpful to present the confidence intervals stratified (for both smokers and nonsmokers) in each cohort.

We agree and have now presented supplementary plots stratified by smoking status, sex, and ancestry.

9) The authors readily conclude that lower AHRR gene body methylation also means higher expression of AHRR; however, they have no functional data to support such a claim, which is not necessarily true given that gene body CpG methylation has been shown to positively correlate with gene expression (for example, see: Ball et al., Nat Biotechnol 2009;27(4):361-8, doi: 10.1038/nbt.1533). This becomes problematic because the whole manuscript is then built on this ungrounded assumption.

We used the iMETHYL database (<http://imethyl.iwate-megabank.org/>) to evaluate whether methylation of *AHRR* CpGs associated with *AHRR* expression in PBMCs, the available tissue that most closely resembles whole blood. Though eQTMs are only listed if they associate at $FDR < .05$, we observed that 2 of the *AHRR* CpGs negatively correlated (table below), supporting our assumption that lower methylation of *AHRR* CpGs associates with higher levels of expression. We have added this information to the Results.

CpG	Position	Gene Name	Beta	pval
cg05575921	chr5:373378	AHRR	-0.0059	1.05E-09
cg25648203	chr5:395444	AHRR	-0.0039	0.016

10) Other aspects of the overall model (as presented on Figure 3) are also too speculative. Why do the authors assume that PTSD cases had more exposure to AhR ligands? Could they provide relevant data to support this claim or at least previous evidence from human cohorts showing regulation of the pathway depending on AhR ligand exposure? Could changes in methylation simply reflect other factors, e.g. differences in genetic background between cases and controls? Likewise, the authors have no data on Th17 or Treg to support differences in immune cell population and function.

Our goal was to offer another plausible mechanism that could induce *AHRR* activation, but we agree that our model was speculative. In this revision, we have simplified this discussion and focused it on what is established in the literature. We also discuss recent papers that have reported *AHRR* methylation in non-smokers exposed to other environmental toxicants. Finally, we acknowledge and discuss that changes in methylation simply reflect other factors, including differences in genetic background between cases and controls in the study limitations.

Reviewer 3:

1. Figure 1 shows significance thresholds for two FDR levels. This doesn't make sense as the FDR adjustment is different for every CpG depending on its ranked p-value.

We remade Figure 1 to reflect the genome-wide threshold for EWAS significance ($P < 3.6 \times 10^{-8}$).

2. P8 L181 "collinearity" isn't really the right term as there is still a good proportion of non-smokers among the cases. I'd suggest that what you have shown is more like an interaction between PTSD and smoking status on AHRR methylation. If one is a smoker there isn't much of an effect of PTSD, whereas if one is a non-smoker then PTSD status has more of an effect.

We removed the term 'collinearity' and changed the text to, "We also evaluated potential differences of effect between PTSD and smoking status by testing the associated CpGs separately in smokers and non-smokers."

3. P10 L217-219 I didn't understand the meaning of "AUC < 1e6" and "AUC >= 1e7). If this is area under the ROC curve then it should be between 0.5 and 1.

This was a typographical error for which we apologize. Cotinine is determined based on arbitrary units (AU) and not an area under the curve (AUC). This has been corrected in the manuscript.

4. Is it assumed that AHRR demethylation is a consequence of smoking - presumably so, would be helpful to clarify. Did you consider whether smoking is a cause or consequence of PTSD? This doesn't have much bearing on the results, except if smoking is a consequence of PTSD and there are unmeasured confounders of the smoking-AHRR association. Then, conditioning/stratifying on smoking could introduce a collider bias that distorts the stratified analysis (such a confounder might be a SNP). I suspect this scenario is somewhat unlikely, especially as the association is stronger in non-smokers, but it might be worth a comment.

We have clarified that issue in the Discussion by adding this phrase: 'Though it is possible that stratifying on smoking could introduce a collider bias that distorts the stratified analysis because of a SNP or other unmeasured confounder, this scenario seems unlikely given that the associations between AHRR CpGs and PTSD are stronger in non-smokers. As such, this cross-sectional analysis is unable to determine whether methylation of AHRR CpGs is likely a cause or consequence of smoking.'

5. Figure S2, some of the p-values are given as <0.001 where the CIs appear to touch or overlap the null - doesn't look right.

Thank you for pointing this out. It was an error in the figure, which has now been corrected.

REVIEWERS' COMMENTS:

Reviewer #1 (Remarks to the Author):

I thank the authors for their detailed responses. I have no major comments, only two minor points.

1. I am not sure that "an epigenetic biomarker of aging" is sufficient detail for a general journal like Nature Comms. I would perhaps mention e.g, how DNAm can be used to predict chronological age with reasonably high accuracy and how deviations from a line of perfect prediction might reflect age acceleration or biological aging.

2. I still think it would be really helpful to run a DNAm-based predictor of smoking e.g.,

<https://pubmed.ncbi.nlm.nih.gov/31466478/>

<https://genomebiology.biomedcentral.com/articles/10.1186/s13059-018-1514-1>

The former will give a prediction of smoking status (current/ex/never) for each participant and the latter will give a more granular smoking score. I appreciate that the authors have investigated some of the lead CpGs from the CHARGE analysis and that both of the references above include some of these sites. However, it would be nice to see what the distribution of predicted smoking looks like in the non-smokers compared to the current smokers. The predictors could also be run with/without the AHRR site to try and reduce complications as mentioned in their responses.

Reviewer #3 (Remarks to the Author):

The authors have addressed my review. I note a few minor issues outstanding

1. The abstract refers to "susceptibility", seeming to imply that DNA methylation may predispose to PTSD onset. But methylation may be the consequence of PTSD, and (appropriately enough) this is not mentioned further in the manuscript.

2. Figure 2 and Figure S4 appear identical, and the description of Figure 2 at line 153 seems incorrect.

3. There are typo/grammar errors on lines 153, 156, 228, 231, 307 and no doubt elsewhere.

4. Reference formatting was garbled between refs 52-54.

Response to Reviewers

Reviewer #1

1. I am not sure that "an epigenetic biomarker of aging" is sufficient detail for a general journal like Nature Comms. I would perhaps mention e.g, how DNAm can be used to predict chronological age with reasonably high accuracy and how deviations from a line of perfect prediction might reflect age acceleration or biological aging.

We agree and have revised that sentence.

2. I still think it would be really helpful to run a DNAm-based predictor of smoking e.g., <https://pubmed.ncbi.nlm.nih.gov/31466478/> <https://genomebiology.biomedcentral.com/articles/10.1186/s13059-018-1514-1>

The former will give a prediction of smoking status (current/ex/never) for each participant and the latter will give a more granular smoking score. I appreciate that the authors have investigated some of the lead CpGs from the CHARGE analysis and that both of the references above include some of these sites. However, it would be nice to see what the distribution of predicted smoking looks like in the non-smokers compared to the current smokers. The predictors could also be run with/without the AHR site to try and reduce complications as mentioned in their responses.

Using the reference provided, we tested the prediction of smoking status with and without the inclusion of cg05575921 in *AHR* in a group of 118 self-identified smokers and 135 nonsmokers from the GTP cohort. The prediction was reasonably consistent with self-reported status. Of the self-identified smokers, 50% were classified as current or former smokers while 50% were classified as never smokers. Of the self-identified non-smokers, only 10% were classified as current or former smokers, with 90% classified as never smokers. However, when cg05575921 was excluded as suggested, the algorithm predicted that all subjects were smokers. Upon further scrutiny of the approach, cg05575921 appears to have the highest weight for predicting the non-smoker class.

Reviewer #3

1. The abstract refers to "susceptibility", seeming to imply that DNA methylation may predispose to PTSD onset. But methylation may be the consequence of PTSD, and (appropriately enough) this is not mentioned further in the manuscript.

We agree and have edited the abstract.

2. Figure 2 and Figure S4 appear identical, and the description of Figure 2 at line 153 seems incorrect.

We apologize for the error, which is now corrected. Figure S4 is the comparison of effect sizes in main model and the smoking-adjusted model. Figure 2 is the overall results for smokers and non-smokers, while Figure S5 is results for each cohort.

3. There are typo/grammar errors on lines 153, 156, 228, 231, 307 and no doubt elsewhere.

We have reviewed the manuscript and corrected typos and grammar errors.

4. Reference formatting was garbled between refs 52-54.

We have corrected the formatting.